

# How long should the fully hillside-closed forest protection be implemented on the Loess Plateau, Shaanxi, China?

Lin Hou[1] and Sijia Hou[2]

[1] College of Forestry, Northwest A&F University, Yangling, Shaanxi, China
[2] College of Transportation, Southeast University, Nanjing, Jiangsu, China

## ABSTRACT

**Background**. Restoration of degraded forest ecosystem is crucial for regional sustainable development. To protect the country's fragile and fragmented environment, the Chinese government initiated an ecological engineering project, the Natural Forest Protection Program, in seventeen provinces in China beginning in 1998. Fully hillside-closed forest protection (vegetation restoration naturally without any artificial disturbance) was one of vital measures of the Natural Forest Protection Program applied nation wide. Whether plant diversity, biomass and age structure of dominant tree species and soil nutrients in protected stands may become better with increase of protected period are still open problems.

**Methods**. We investigated community diversity, biomass of dominant tree species, age structures, and analyzed soil chemical properties of a *Pinus tabulaeformis* population at protected sites representing different protected ages at Huanglongshan Forest Bureau on the Loess Plateau, Shaanxi, China.

**Results**. Plant species richness of *Pinus tabulaeformis* community was significantly affected ($p < 0.05$) by forest protection and the effect attenuated with protection age. Shannon evenness index of plant species generally increased with protection age. Stands protected for 45 years had the highest tree biomass and considerable natural regeneration capacity. Contents of organic carbon, available phosphorus and available potassium in top soil increased in protected stands less than 45 years, however decreased significantly thereafter. Long-term forest protection also decreased the content of mineral nitrogen in top soil.

**Discussion**. We found that the richness of shrubs and herbs was significantly affected by forest protection, and evenness indices of tree, shrub and herb increased inconsistently with protected ages. Forest protection created more complex age structures and tree densities with increasing age of protection. Content of soil mineral nitrogen at 0–20 cm soil depth showed a decreasing trend in stands of up to 30 years. Soil available phosphorus and potassium contents were higher in stands with greater proportions of big and medium trees. Long-term protection (>45 years) of *Pinus tabulaeformis* stands in southeast Loess Plateau, China, may be associated with decreasing plant species richness, proportion of medium to large trees, dominant biomass of *Pinus tabulaeformis* and soil nutrients.

Corresponding author
Lin Hou, houlin1969@163.com

## INTRODUCTION

Ecological restoration is being recognized as an international priority (*Aronson & Alexander, 2013*; *Wortley, Hero & Howes, 2013*) and it plays a crucial role in rebuilding ecological equilibrium and reversing ecosystem degradation (*Ma, Lv & Li, 2013*). As a part of ecological engineering (*Mitsch, 2012*), the practice is being widely incorporated into natural resource strategies from the local to global level (*Wortley, Hero & Howes, 2013*).

To protect the country's fragile and fragmented environment, the Chinese government initiated an ecological engineering project, the Natural Forest Protection Program (NFPP), beginning in 1998 (*Xu et al., 2006*). Logging and harvesting of partial or full timber was prohibited in protected areas from 1998 to 2008 (*Xu et al., 2006*). Fully hillside-closed forest protection (vegetation restoration naturally without any artificial disturbance) was applied nation-wide. Ecosystems have the capacity to self-organize and the self-design or self-organizational properties of natural systems is an essential component to ecological engineering (*Bergen, Bolton & Fridley, 2001*). Obviously, fully hillside-closed forest protection is in accord with the ecological engineering principle self-design.

The previous studies regarding NFPP have mainly focused upon the introduction of the related policy issues, the spatial-temporal succession of regional vegetation (*Huang et al., 2014*) and ecological restoration programs and payments (*Yin & Zhao, 2012*). However, a range of questions remain, particularly in relation to stand function and associated environmental parameters following stand protection. We hypothesize that fully hillside-closed forest protection may promote plant diversity, biomass and age structure of dominant tree species and soil nutrients with increase of protected period. The objectives of this study are to address a few of these key knowledge gaps, including: (i) do the stands exhibit significant differences in plant assemblage; (ii) does soil fertility change with stand age structure; (iii) can a functional relationship be defined regarding length of stand protection and stand quality, i.e., are stands protected for longer time frames ''better'' than other stands; and (iv) based on findings of i–iii above, can a preliminary estimate regarding the optimal time span for *Pinus tabulaeformis* stands be recommended to the Natural Forest Protection Program?

## MATERIALS AND METHODS

### Site description

The study was conducted in Huanglong County (35°28′49″–36°02′01″N, 109°38′49″– 110°12′47″E) on the southeast Loess Plateau, Shaanxi, China. Stands in this area (a part of NFPP area) play key ecological roles in abatement of soil erosion and mitigation of sand storm (*Chen et al., 2014*). The vegetation type is a northern deciduous broad-leaved forest sub-region. *Pinus tabulaeformis* is dominant tree species in the currently existing stands. The associated tree species are *Quercus liaotungensis*, *Syringa oblate*, *Populus davidiana*, *Prunus davidiana*, *Betula platyphylla* and *Toxicodendron vernicifluum*. Shrubs and herb species in the understory are abundant. The altitude ranges from 1,100 to 1,300 m. It is dominated by a warm temperate and semi-humid continental climate. The annual average precipitation is 612 mm and the mean atmospheric temperature is 8.6 °C. Cinnamon soil

is the main soil type in the forest region. Due to poor communication and a small human population in the past years, stands on some special sites have not been disturbed since 1950, especially since 1998. We consulted forest resource archive data of Huanglongshan Forest Bureau, Yanan, Shaanxi, China to find the year of forest protection and chose plots from forest farms (Fig. 1). According to the data, stands with protected age sequences were found in four forest farms (Table 1).

## Field methods

We chose plots randomly. To find typical stand plots in the same protected age, not only similar altitude and canopy density, but also various directions, gradients, positions and density of dominant tree species were considered as principles. The field investigation and sampling was conducted between June 5 and July 15, 2003. Each plot of trees, shrubs and herbs was 20 m × 20 m, 2 m × 2 m and 1 m × 1 m respectively. Five sub-plots of shrubs, herbs and regenerating seedlings were arrayed diagonally in each tree plot respectively (Fig. 2). The indices, species, number, Height ($H$), diameter at breast height (DBH) and canopy density of trees, and species, height, cover ratio, number of shrubs, herbs and regenerating seedlings were measured. All community data were collected from 27 tree plots spreading among the age cohorts and 270 sub-plots (Table 1). Three soil samples were obtained randomly by a special drill in each tree plot. Surface soils (0–20 cm depth) at all sites were assessed for soil properties including organic carbon, mineral nitrogen, available phosphorous and potassium.

## Community diversity

The diversity–productivity relationship (DPR) has been paid much attention during the past two decades (*Hooper et al., 2005*). In most DPR studies, richness has been chosen as the index of species diversity to define and interpret DPRs (*Zhang, Chen & Reich, 2012*). However, the simplex index cannot completely represent species diversity (*Bock, Jones & Bock, 2007*) in relation to ecosystem functioning for it ignores the effect of species evenness on interspecific interactions (*Hillebrand, Bennett & Cadotte, 2008*; *Kirwan et al., 2007*; *Turnbull & Hector, 2010*). Based on 54 studies, the importance of species richness and evenness in influencing diversity-associated productivity has been demonstrated in a meta-analysis (*Zhang, Chen & Reich, 2012*). In this study, we chose indices of richness and evenness to reflect characteristics of community. Species richness index ($S$) was derived from field survey data. To characterize the diversity of the stand community, the Shannon–Wiener index ($H'$) and evenness index ($J'$) were calculated as the following:

Shannon–Wiener index $H' = -\sum P_i \ln P_i$

Shannon evenness index $J' = \dfrac{H'}{\ln S}$

where $P_i$ is the relative frequency of the ith species, and $S$ is total number of species in plots and subplots (*Magurran, 2004*).

## Biomass of dominant tree species

Average DBH (cm) and height (m) of *Pinus tabulaeformis* in each plot were calculated and living biomass (Mgha$^{-1}$) of whole trees (*Pinus tabulaeformis*) were estimated according to

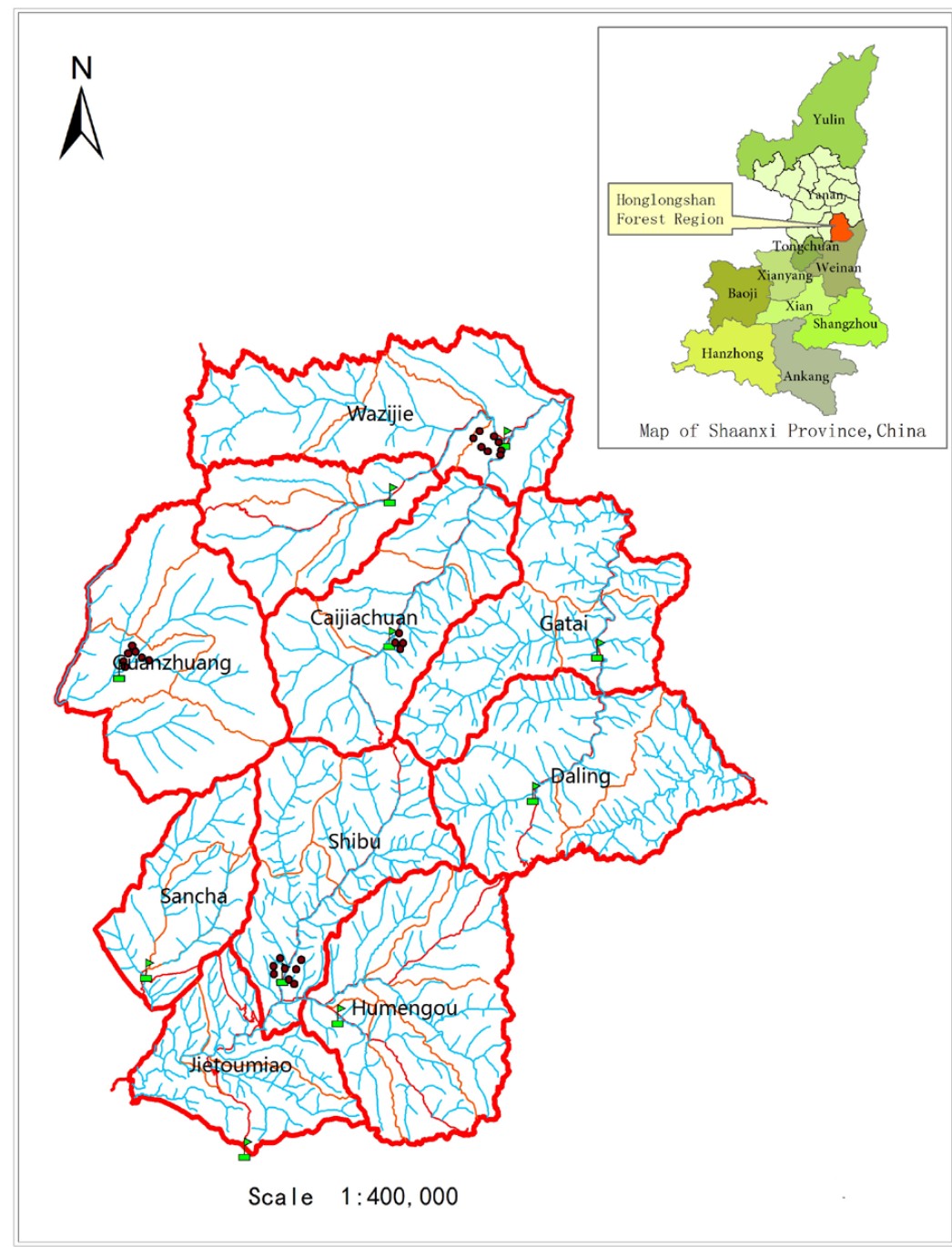

**Figure 1  Position of plots.** Dots on the figure were plots chosen in each forest farm (Map credit: Huang-longshan Forest Bureau, Yanan, Shaanxi, China).

**Table 1  General information of plots.**

| Forest farm | Plot no. | Altitude (m) | Direction | Gradient (°) | Position | Density of dominant tree species (trees ha⁻¹) | Canopy density | Protecting age (a) |
|---|---|---|---|---|---|---|---|---|
| | 1 | 1,170 | Southeast | 27 | Upper | 1,025 | 0.30 | |
| | 2 | 1,150 | Southeast | 29 | Middle | 1,075 | 0.35 | |
| Shibu | 3 | 1,165 | Northeast | 24 | Middle | 1,075 | 0.30 | 16 |
| | 4 | 1,135 | Northeast | 26 | Lower | 1,100 | 0.30 | |
| | 5 | 1,295 | North | 19 | Middle | 1,050 | 0.30 | |
| | 6 | 1,154 | North | 22.3 | Middle | 1,050 | 0.60 | |
| | 7 | 1,167 | Northwest | 24 | Lower | 675 | 0.50 | |
| Guanzhuang | 8 | 1,180 | South | 35 | Upper | 700 | 0.70 | 30 |
| | 9 | 1,165 | South | 35 | Lower | 1,350 | 0.60 | |
| | 10 | 1,180 | South | 22 | Upper | 1,375 | 0.60 | |
| | 11 | 1,163 | South | 25 | Upper | 1,050 | 0.50 | |
| | 12 | 1,170 | North | 24 | Upper | 750 | 0.60 | |
| | 13 | 1,160 | North | 22.3 | Middle | 800 | 0.70 | |
| | 14 | 1,175 | North | 21 | Upper | 625 | 0.60 | |
| Wazijie | 15 | 1,163 | North | 23 | Middle | 650 | 0.70 | 45 |
| | 16 | 1,154 | North | 26 | Lower | 600 | 0.70 | |
| | 17 | 1,120 | | | Gully bottom | 730 | 0.70 | |
| | 18 | 1,130 | North | 10 | Lower | 760 | 0.60 | |
| | 19 | 1,150 | North | 8 | Ridge top | 640 | 0.70 | |
| | 20 | 1,200 | Northeast | 19 | Middle | 1525 | 0.60 | |
| | 21 | 1,155 | Northern | 5 | Ridge top | 1,550 | 0.40 | 60 |
| | 22 | 1,150 | North | 18 | Upper | 1,725 | 0.40 | |
| Caijiachuan | 23 | 1,130 | North | 16 | Lower | 1,475 | 0.40 | |
| | 24 | 1,205 | Northeast | 10 | Middle | 1,900 | 0.30 | |
| | 25 | 1,200 | Northeast | 5 | Lower | 1,700 | 0.30 | 75 |
| | 26 | 1,185 | | | Mesa | 1,425 | 0.30 | |
| | 27 | 1,135 | North | 18 | Middle | 1,400 | 0.40 | |

the literature (*Chen & Peng, 1996*; *Pan et al., 2004*).

$$Y = 15.525 + 0.6269v$$

$$\ln v = 0.99138 \ln \left(D^2 H\right) - 10.30211$$

where $Y$ is the living biomass of trees (Mgha⁻¹), $v$ is the stand growing stock (m³ ha⁻¹), $D$ (cm) is diameter at breast height and $H$ (m) is height.

Combining the density of dominant tree species (Table 1) with equations, the biomass of *Pinus tabulaeformis* in protected stands was determined.

## Age structures

DBH of tree species is correlated significantly to their ages under the same environmental condition (*Parker & Peet, 1984*). Lacking analytic wood data, we adopted DBH structures of *Pinus tabulaeformis* population instead of its age structures. Combining DBH and $H$,

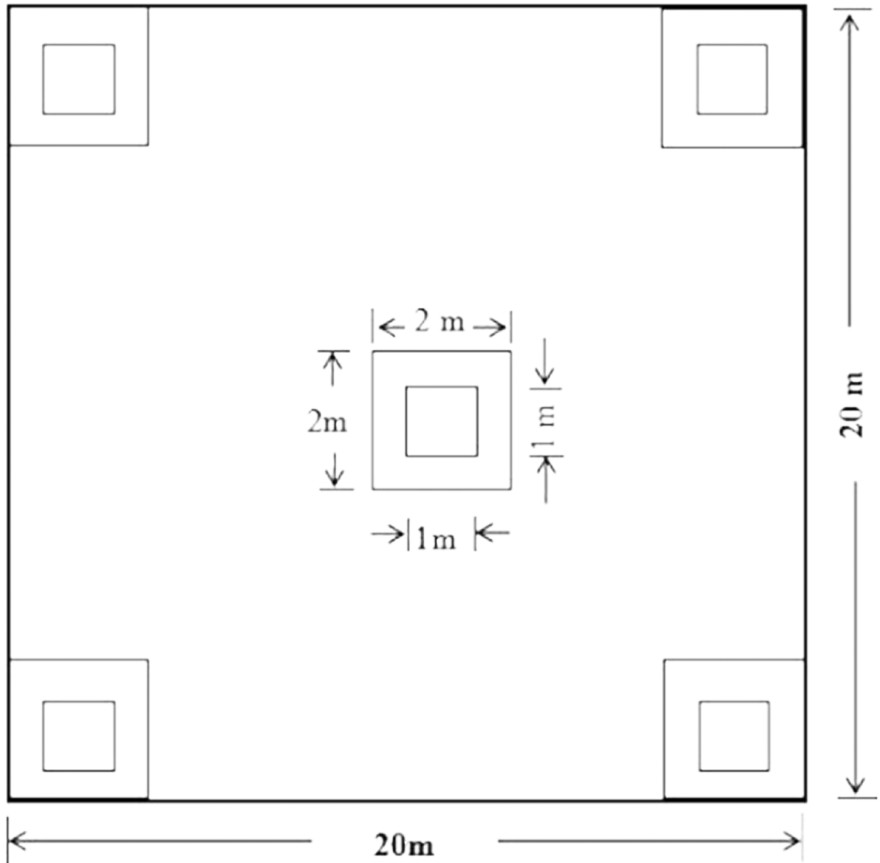

**Figure 2  Layout of subplots.** The plots 20 m × 20 m, the subplots 2 m × 2 m and 1 m × 1 m were used for the investigation of trees, shrubs and herbs respectively.

age structures of *Pinus tabulaeformis* population were classified as following: I seedling, $H \leq 0.30$ m; II young tree, $0.30$ m $< H \leq 2.00$ m, DBH $\leq 6.00$ cm; III small tree, $H > 2.0$ m, $6.0$ cm $<$ DBH $\leq 12.0$ cm; IV medium tree, $12.0$ cm $<$ DBH $\leq 20.0$ cm; V big tree, DBH $> 20.0$ cm. The ratio of seedlings, young trees, small trees, medium trees and big trees in stands with the same protection age was used to illustrate age structures. The probable age of an individual was determined by their whorled branches.

## Chemical analyses

Analyses were made on air-dry soil material that passed through a 2 mm sieve. Soil organic carbon content (SOC) was determined by dry combustion with a TOC/TON analyzer (TOC-VTH-2000A; Shimadzu Corporation, Japan). Soil mineral nitrogen (ammonium nitrogen, $NH^+_4-N$ and nitric nitrogen, $NO_3^--N$) content was determined by the colorimetric method with automatic flow injection (AA3, BRAN + LUEBBE; Germany). Available phosphorus content was extracted in 0.5M $NaHCO_3$ and determined by Mo-Sb colorimetry. Available potassium content was determined by method of flame photometry (*Bao, 2000*).

**Table 2  Richness of plant species in protected stands.**

| Resoration age(a) | Tree | Shrub | Herb |
|---|---|---|---|
| 16 | Pinus tabulaeformis<br>Populus davidiana<br>Syringa oblata | Lespedeza dahurica<br>Lespedeza floribunda<br>Rosa hugonis<br>Sophora viciifolia | Astragalus kifonsanicus<br>Artemisia mongolica<br>Artemisia giraldii<br>Aster tataricus<br>Bothriochloa ischaemum<br>Bupleurum chinense<br>Kengia serotina<br>Lamium barbatum<br>Patrinia heterophylla<br>Rhaponticum uniflorum<br>Scutellaria baicalensis<br>Viola chaerophylloides |
| 30 | Betula platyphylla<br>Conus walteri wanger<br>Pinus tabulaeformis<br>Prunus davidiana<br>Prunus tomenosa<br>Xanthoceras sorbifolia<br>Quercus Liaotungensis<br>Syringa oblata | Acer ginnala<br>Berberis dielsiana<br>Clematis brevicaudata<br>Clematis fruticosa<br>Cotoneaster multiflorus<br>Lespedeza dahurica<br>Lonicera ferdinandii<br>Ostryopsis davidiana<br>Periploca sepium<br>Rhamnus davurica<br>Rhamnus utilis<br>Rosa hugonis<br>Rubus corchorifolius<br>Sophora viciifolia<br>Spiraea fritschiana<br>Ziziphus jujube var.spinosus | Adenophora potaninii<br>Adenophora stricta<br>Agrimonia pilosa<br>Anaphalis margaritacea<br>Artemisia giraldii<br>Artemisia gmelinii<br>Artemisia mongolica<br>Aster tataricus<br>Bothriochloa ischaemum<br>Bupleurum chinense<br>Carpesium divaricatum<br>Discorea nippnica<br>Gentiana macrophylla<br>Kengia serotina<br>Leontopodium leontopodioides<br>Lilium pumilum<br>Lysimachia barystachys<br>Melissitus ruthenicus<br>Patrinia heterophylla<br>Pennisetum clandestinum<br>Polygonatum odoratum<br>Potentilla supina<br>Sanguisorba officinalis<br>Saussurea morifolia<br>Saussurea nivea<br>Saussurea petrovii<br>Saussurea salsa<br>Scutellaria baicalensis<br>Spodiopogon sibiricus |

**Table 2** (*continued*)

| Resoration age(a) | Tree | Shrub | Herb |
|---|---|---|---|
| | | | *Thalictrum prezewalskii* |
| | | | *Urena lobata* |
| | | | *Vicia unijuga* |
| | | | *Viola chaerophylloides* |
| | | | *Viola selkirkii* |
| | | | *Viola yedoensis* |
| 45 | *Betula platyphylla* | *Acer ginnala* | *Agrimonia pilosa* |
| | *Pinus tabulaeformis* | *Lespedeza dahurica* | *Anaphalis margaritacea* |
| | *Quercus Liaotungensis* | *Lonicera maccki* | *Artemisia mongolica* |
| | *Syringa oblata* | *Rubus corchorifolius* | *Aster tataricus* |
| | | *Spiraea fritschiana* | *Bothriochloa ischaemum* |
| | | | *Kengia serotina* |
| | | | *Neottianthe cucullata* |
| | | | *Potentilla discolor* |
| | | | *Spodiopogon sibiricus* |
| | | | *Urena lobata* |
| | | | *Viola chaerophylloides* |
| | | | *Viola japonica var. stenopetala* |
| | | | *Viola selkirkii* |
| | | | *Viola yedoensis* |
| 60 | *Pinus tabulaeformis* | *Acer ginnala* | *Adenophora stricta* |
| | *Populus davidiana* | *Berberis dolichobotrys* | *Anaphalis margaritacea* |
| | *Prunus tomenosa* | *Cotoneaster zbakelii* | *Artemisia gmelinii* |
| | *Quercus Liaotungensis* | *Lespedeza dahurica* | *Artemisia mongolica* |
| | *Syringa oblata* | *Ostryopsis davidiana* | *Aster tataricus* |
| | *Toxicodendron vernicifluum* | *Rosa hugonis* | *Bothriochloa ischaemum* |
| | | *Rubus corchorifolius* | *Bupleurum chinense* |
| | | *Spiraea fritschiana* | *Kengia serotina* |
| | | | *Neottianthe cucullata* |
| | | | *Polygonatum sibircum* |
| | | | *Potentilla discolor* |
| | | | *Scutellaria baicalensis* |
| | | | *Spodiopogon sibiricus* |
| | | | *Vicia cracca* |
| | | | *Viola chaerophylloides* |
| | | | *Viola selkirkii* |
| | | | *Urena lobata* |

| Resoration age(a) | Tree | Shrub | Herb |
|---|---|---|---|
| 75 | *Pinus tabulaeformis* | *Acer ginnala* | *Anaphalis margaritacea* |
| | *Populus davidiana* | *Clematis fruticosa* | *Artemisia mongolica* |
| | *Quercus Liaotungensis* | *Cotoneaster zbakelii* | *Aster tataricus* |
| | *Syringa oblata* | *Indigofera amblyantha* | *Bothriochloa ischaemum* |
| | *Toxicodendron vernicifluum* | *Lespedeza dahurica* | *Kengia serotina* |
| | | *Lonicera maccki* | *Sanguisorba officinalis* |
| | | *Ostryopsis davidiana* | *Spodiopogon sibiricus* |
| | | *Rubus corchorifolius* | *Thalictrum prezewalskii* |
| | | *Spiraea fritschiana* | *Urena lobata* |
| | | | *Viola chaerophylloides* |
| | | | *Vicia unijuga* |
| | | | *Viola selkirkii* |

## Data processing and analysis

SPSS 17.0 and Origin8.0 (OriginLab Corporation) software were used for statistical analysis and plotting. Histograms were performed by SPSS 17.0 based on the residual distribution. We hypothesized the variance was homogeneous firstly. Then, homogeneity test of variance was performed by SPSS 17.0. If our hypothesis was true, one-way analysis of variance (ANOVA) following Fisher's least significant difference (LSD) test ($p < 0.05$) was used to compare the protection age effects on diversity of plant community and soil nutrients respectively.

## RESULTS

### Diversity of plants in protected stands

Data from 27 plots (Table 1) representing protection age of stands and richness of trees, shrubs and herbs (Table 2) were compared. Richness index of trees (8), shrubs (17) and herbs (35) was highest in the stand protected for 30 years (Table 2). A significant difference in the tree species richness index was observed in the 30 year protected stand compared to stands protected for 16 years ($n = 11$, $p = 0.000$) and 45 years ($n = 14$, $p = 0.000$), but not in other stands with different protected ages (Fig. 3). The richness index of within stand shrubs differed significantly between stands protected for 30 years compared to stands protected for 16 years ($n = 55$, $p = 0.000$), 45 years ($n = 65$, $p = 0.000$), 60 years ($n = 50$, $p = 0.000$) and 75 years ($n = 50$, $p = 0.000$) (Fig. 3). Significant differences in the within-stand herb richness index were also found in stands protected (i) 16 years and 60, 75 years, (ii) 45 years and 60 years, 75 years (Fig. 3). The within stand herb richness index in the stands protected for 30 years differed significantly from stands protected for 16 years ($n = 55$, $p = 0.000$), 45 years ($n = 65$, $p = 0.000$), 60 years ($n = 50$, $p = 0.000$) and 75 years ($n = 50$, $p = 0.000$) (Fig. 3). Richness of within stand herb at stands protected for 16 years also varied significantly from stands protected for 60 years and 75 years (Fig. 3).

Shannon-Wiener evenness index of tree, shrub and herb was highest in stands protected for 30 years, 45 years and 75 years respectively (Fig. 4A). The index of herb generally increased with protected ages except in stands protected for 16 years to 30 years (Fig. 4A).

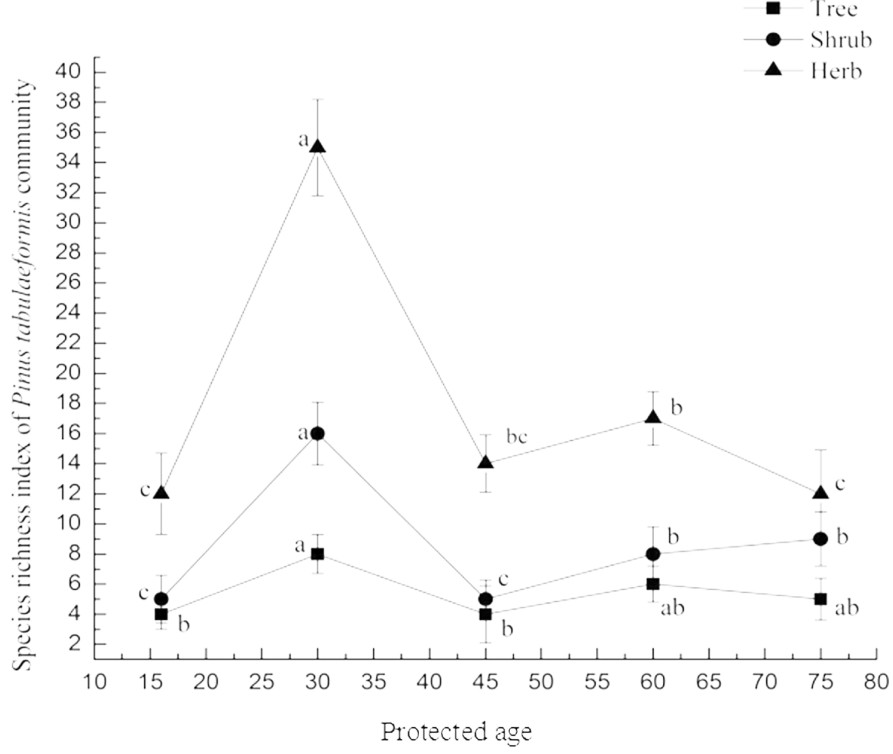

**Figure 3   Richness of *Pinus tabulaeformis* community in protected stands.** The values are the mean
± SD. Different letters in the same layer in the figure indicate significant differences between groups based
on LSD ($p < 0.05$). Confidence interval of 95% for richness of tress, shrubs and herbs among protection
years is [3.32,7.48], [2.03,14.37] and [5.93,30.07] respectively.

However, the index of tree and shrub fluctuated with stand protected years and did not
follow a trending relationship (Fig. 4A). Tree and shrub Shannon-Wiener index increased
with stand protection age, with the exception of tree index 30–45 year stand protection
and shrub index 45–60 year stand protection (Fig. 4A). To clearly show the trend of
evenness index with age, linear fit of trees, shrubs and herbs was given out by Origin8.0
(OriginLab Corporation). With protection year increase, species distribution tended to be
more homogeneous (Fig. 4B).

## Biomass of *Pinus tabulaeformis* in protected stands

Biomass of *Pinus tabulaeformis* increased in stands until 45 years of forest protection;
however, for sites older than this protection age, stand biomass decreased (Fig. 5). Peak
biomass was $70.60 \pm 8.00$ t ha$^{-1}$ in the stand protected for 45 years, while biomass in the
stand protected for 75 years ($19.90 \pm 9.20$ t ha$^{-1}$) was lower than the stand protected for
16 years ($23.70 \pm 17.10$ t ha$^{-1}$) (Fig. 5).

## Age structure of *Pinus tabulaeformis* population in protected stands

Although age classes of *Pinus tabulaeformis* occurred in protected stands, they varied greatly
(Fig. 6). Only young (II) and small trees (III) were found in the stand protected for 16 years,
small (III) and medium trees (IV) dominated the stand protected for 30 years (Fig. 6). For

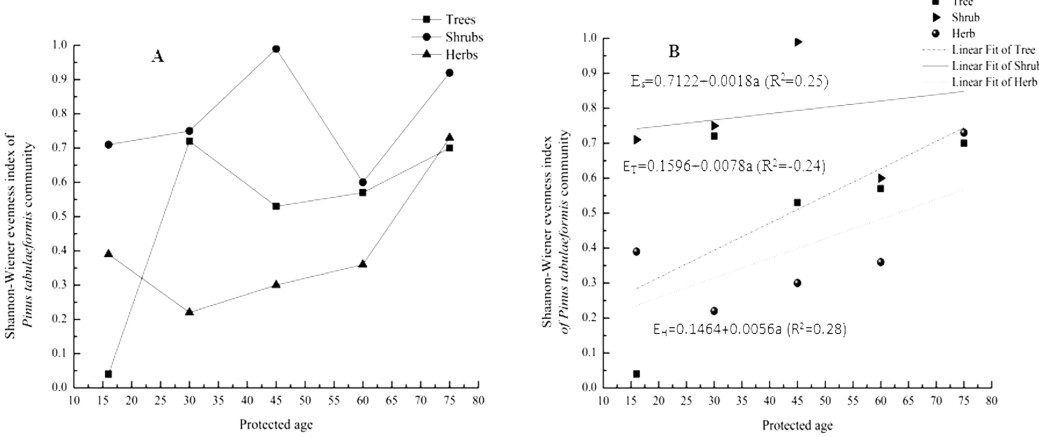

**Figure 4  Shannon–Wiener evenness index of *Pinus tabulaeformis* community in protected stands.**
Letters ET, ES, EH and a in Fig. 4B stand for the evenness index of tree, shrub, herb and protection year respectively. Confidence interval of 95% for evenness of tress, shrubs and herbs is [0.16, 0.85], [0.60, 0.99] and [0.16, 0.64] respectively. Equations in Fig. 4B show the relationship between evenness index and protected year of stands. $E_T$, $E_s$, $E_H$ show a represented evenness index of tree, shrub, herb and protected year of stands, respectively.

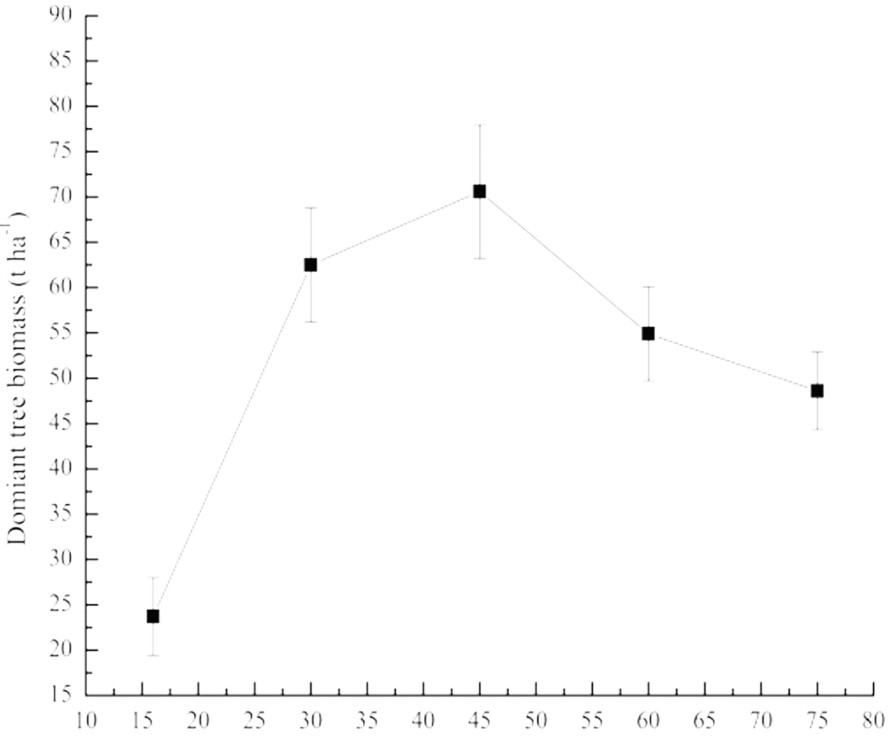

**Figure 5  Biomass of *Pinus tabulaeformis* in protected stands.** Error bars are from one-way analysis of variance (ANOVA). Confidence interval of 95% for biomass of dominant tress is [22.87, 74.25].

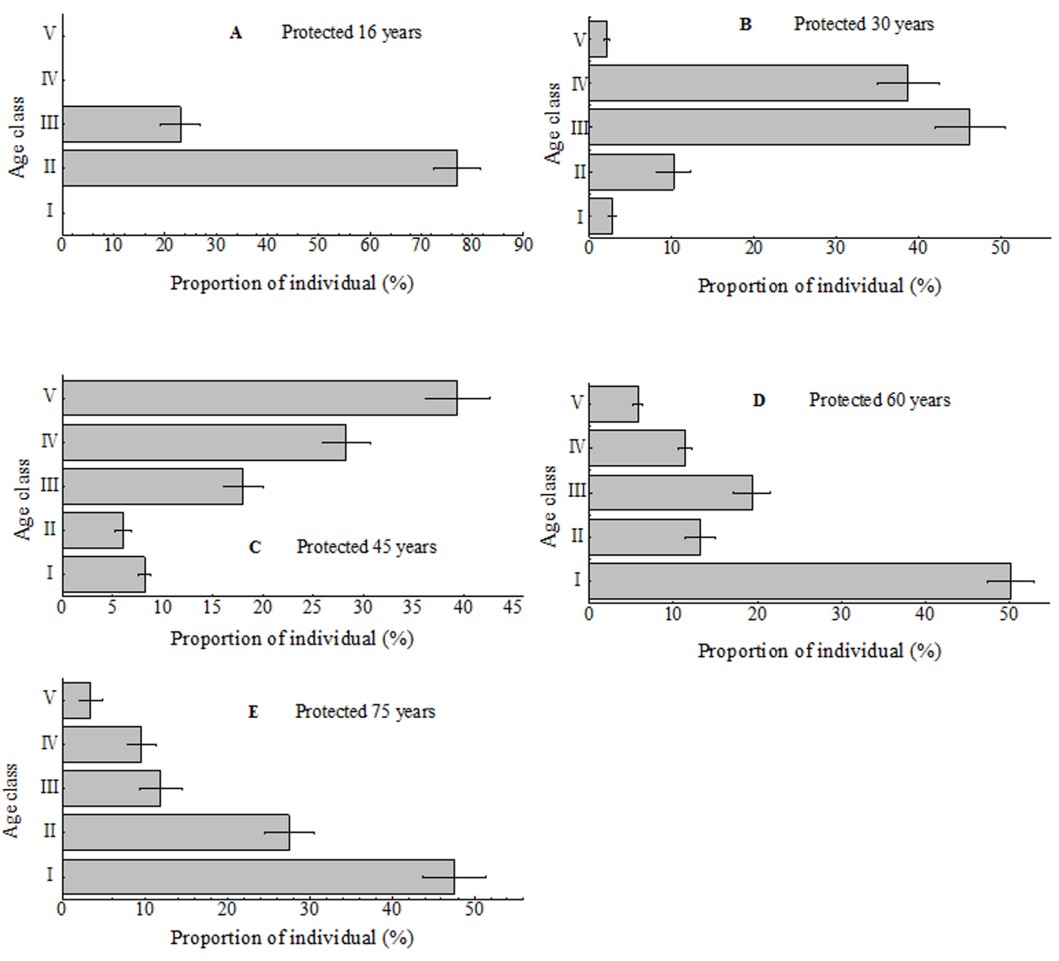

**Figure 6** **Age structure of *Pinus tabulaeformis* population in protected stands.** The Roman numerals (I–V) in figures stand for age structures of *Pinus tabulaeformis* population. I seedling, $H \leq 0.30$ m; II young tree, 0.30 m 2.0 m, 6.0 cm < DBH $\leq$ 12.0 cm; IV medium tree, 12.0 cm < DBH $\leq$ 20.0 cm; V big tree, DBH > 20.0 cm.

the stand protected for 45 years, big (V) and medium trees (IV) were main components, but seedlings (I) and young trees were considerable also (Fig. 6). In contrast, for stands protected for 60 and 75 years, seedlings (I) were the dominant component, followed by young (II) and small trees (III), with big trees (V) lowest in distribution (Fig. 6).

## Soil nutrients

Significant differences of soil organic carbon content at 0–20 cm soil depth were observed between the stands, with higher soil organic carbon content observed in stands protected for longer than 30 years (Fig. 7A). Content of mineral nitrogen at 0–20 cm soil depth demonstrated significant differences in stands before and after the protected 30 years (Fig. 7B). No significant differences were found between stands protected for 16 years and 30 years, and among stands after 30 years (Fig. 7B). Content of available phosphorus at 0–20 cm soil depth increased as protection of stand age increased, with significant differences observed mostly at youngest and oldest stand ages (Fig. 7C). No significant

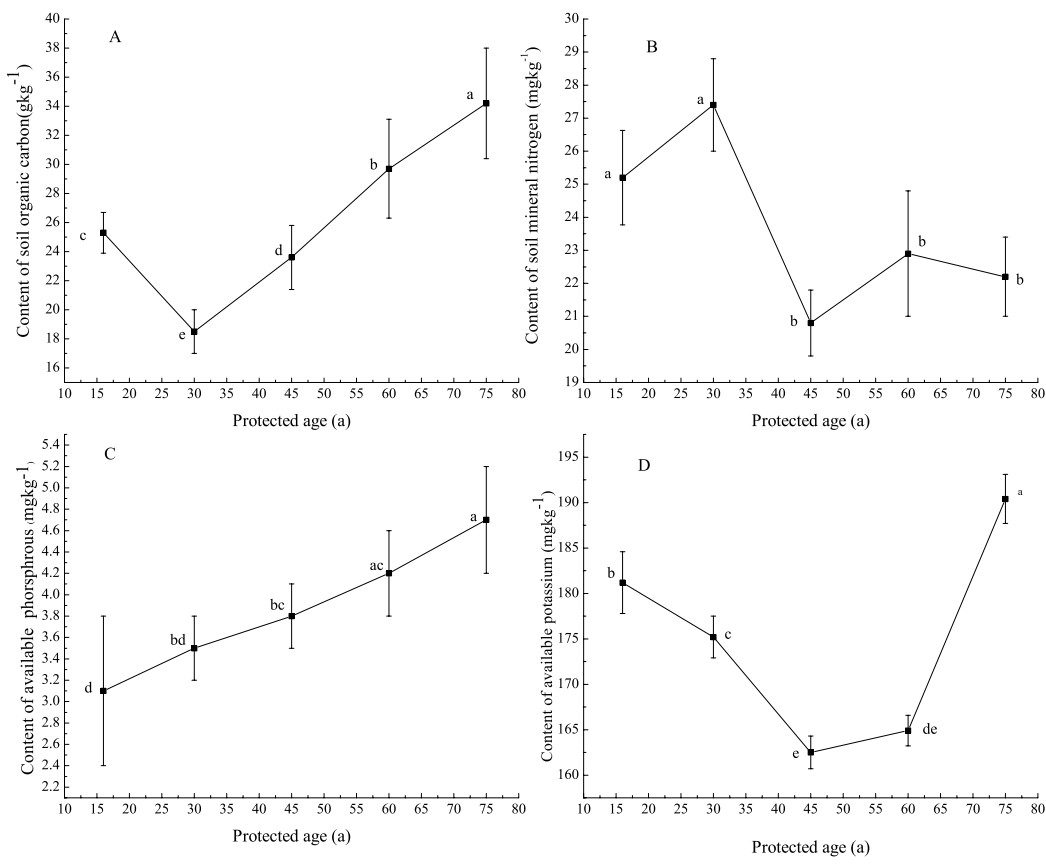

**Figure 7 Soil nutrients in protected stands.** The subfigures (A), (B), (C) and (D) demonstrate the dynamics of soil organic carbon (gkg$^{-1}$), soil mineral nitrogen (mgkg$^{-1}$), soil available phosphorus (mgkg$^{-1}$) and soil available potassium (mgkg$^{-1}$) with protection of stands, respectively. Disparate lowercase letters on the figure represent significant difference ($p < 0.05$).

difference in available phosphorus was observed in stands between 30 and 45 years of protection, and between 45 and 60 years of protection (Fig. 7C). Content of available potassium at 0–20 cm soil depth decreased in stands younger than 45 years and thereafter increased (Fig. 7D). Significant differences were demonstrated among stands with different protection ages, except at ages 45 and 60 (Fig. 7D).

# DISCUSSION

## Response of plant diversity to forest protection

Species richness is one measure of biodiversity and is very important for ecosystem functioning, stability and integrity (*Coroi et al., 2004*). We found that richness of shrubs and herbs was significantly affected by forest protection, although richness (Fig. 3) and evenness (Fig. 4) indices of tree, shrub and herb increased in an unpredictable manner with protected age. The richness of plant species increased in stands protected for 16 years (species numbers of tree, shrub and herb are 3, 4 and 12 respectively) to 30 years (species numbers of tree, shrub and herb are 8, 16 and 35 respectively), decreased in stands protected for 30 years to 45 years (species numbers of tree, shrub and herb are 4, 5 and

14 respectively) and remained fairly stable in stands protected for longer than 45 years (species numbers of tree, shrub and herb are 6, 8 and 17 respectively in stands protected for 60 years; species numbers of tree, shrub and herb are 5, 9 and 12 respectively in stands protected for 75 years) (Table 2). Due to adequate sunlight and growing spaces, some pioneer tree species (*Populus davidiana*, *Betula platyphylla*) and drought resistant shrubs (*Rubus corchorifolius*, *Rosa hugonis*, etc.) and herbs (*Artemisia gmelinii*, *Saussurea petrovii*, etc.) were more prevalent in the younger forest protection sites, increasing plant species richness of these stands (Table 2). With the growth of trees, canopy density increased and some drought resistant plant species disappeared. Advance regeneration seedlings in stands protected for 45 years and older made up a large proportion of the species observed, impeding invasive plant species and stabilizing plant diversity of the community assemblage. Inherent spatial variability within the landscape may provide a possible explanation for this pattern observed, since *Pinus tabulaeformis* stands are distributed across variable site conditions within the region. Soil moisture is considered to be the key limiting factor on the Loess Plateau for differences in plant species growth and regeneration (*Chen et al., 2014*) and it is possible that the differences in soil property as observed in this study affected plant-available moisture.

Forest protection in Huanglongshan forest region, Yanan, Shaanxi, China was initiated in 1950 from forest resources archives. Stand structure within the protection area under the natural restoration condition differed. Stands with diversified age structure were richer in species than stands with less diversified structure (*Thompson, 2012*). Findings in this study partly support this notion. Stands protected for greater than 16 years had more species with diverse age structures and plant species richness (Fig. 3). Age class structure in stands protected for 30 years were generally simpler than stands protected for longer periods (Fig. 6). However, stands older than 30 years of protection had lower richness index of tress and understory species (Fig. 4).

Our results suggest that sustainable forest protection can potentially contribute to plant diversity conservation by increasing species richness generally (Table 2) and promoting even distribution of trees and herbs (Fig. 4).

## Response of age structure to forest protection

Forest protection created more complex age structures (Fig. 5) and tree densities with increasing age of protection (Table 1). Seedlings, medium and big trees were absent in younger stands (Fig. 6) which indicated tree biomass was low (Fig. 5) and lacked natural regeneration capacity. Although plants species were most abundant in stands protected for 30 years (Fig. 3), this protection age contained the lowest proportion of big trees (V) among age classes (Fig. 6) limiting tree biomass. Both seedlings (Fig. 6) and density of trees (Fig. 3) in older (>60 years) protected stands were higher than in younger stands, suggesting a better natural regeneration capacity. However, more seedlings and small trees without adequate big trees (Fig. 6) in some older stands were evidence of insufficient productivity of such stands (Fig. 5).

Our results support the widely accepted view that the rate of stand biomass accumulation peaks in the early stage of development, usually at the time of canopy closure, and declines

thereafter (*Acker et al., 2002*; *McMahon & Schlesinger, 2010*; *Sarah Lesley Taylor, 2005*; *Xu et al., 2012*). The stand protected for 45 years had not only the highest canopy density (Table 1), but also the highest proportion of big trees and tree biomass as well as considerable seedling density (Fig. 6), suggesting adequate regeneration capacity at this age.

## Response of soil nutrients to forest protection

Vegetation plays a key role in maintaining the soils in which they grow (*Mishra, Sharma & Khan, 2003*), by directly influencing soil nutrients accumulation and consequently soil development via above ground inputs (*Blazejewski et al., 2009*; *Drouin et al., 2011*; *Giese et al., 2000*). Litter fall and its decomposition is an important mechanism governing soil chemical properties (*Mishra, Sharma & Khan, 2003*), especially the upper soil layer (*Ma et al., 2007*).

In the present study, *Pinus tabulaeformis* tree growth (Fig. 5) and understory plant species richness increased quickly for stands protected less than 30 years (Fig. 3), however litter input to soil was lower due to the absence of big trees in these stands (Fig. 6). Tree and canopy density (Table 1) decreased in stands protected for more than 30 years, with highest values observed in stands protected for 45 years (Fig. 5). Increased litter input, decomposition rate and higher soil organic carbon contents were also observed at older forest sites (Fig. 7A).

Content of soil mineral nitrogen at 0–20 cm soil depth showed a decreasing trend in stands of up to 30 years of protection although no significant differences were found among stands (Fig. 7B). This trend does not support previous studies which have observed that young or developing stands accumulate forest floor nitrogen, tending towards relatively stable conditions in undisturbed mature forests.

The primary source of phosphorus and potassium in terrestrial ecosystems are derived from mineral materials in weathering parent rock (*Filippelli, 2008*; *Sheng, 2005*; *Smeck, 1985*; *Tiessen, Stewart & Cole, 1984*). A proportion of the released phosphorus and potassium, available in exchangeable and soluble (available) fractions, can be assimilated by plants and soil microorganisms directly (*Schachtman, Reid & Ayling, 1998*; *Sheng, 2005*). Soil phosphorus availability is also enhanced through phosphorus solubilizing and mineralizing microbial biomass (*Richardson & Simpson, 2011*). Many soil microorganisms excrete organic acids to directly dissolve rock potassium to bring the potassium into solution (*Bennett, 1998*; *Friedrich et al., 1991*; *Groudev, 2010*; *Ullman et al., 1996*).

In the present study, soil available phosphorus (Fig. 7C) and potassium (Fig. 7D) contents were higher in stands with greater proportions of big and medium trees. We suggest that the stands with greater biomass accumulated more litter and humic mineral in the top soil, which provided a substantial energy source and favorable conditions for microbial activity (*Fontaine, Mariotti & Abbadie, 2003*). In younger stands, more nutrients may be taken up by the vegetation during intense tree growth phase than can be replaced within the soil from mineral weathering and litter decomposition (*Brais, Camiré & Paré, 1995*) which may explain why soil available potassium decreased in stands of up to 45 years of protection in this study (Fig. 7D).

### The optimal age for the fully hillside-closed forest protection

No restoration project is undertaken in a social vacuum (*Knight et al., 2010*). The goods and services provided by forests are an important source of income for local people in the rural part of China (*Ma, Lv & Li, 2013*). Even when the intentions of ecological restoration are good and the restoration strategy suitable for the environmental conditions (*Ma, Lv & Li, 2013*), restoration action will not be sustainable if it does not take into account the profit potential of local people.

Our results showed that long-term protection (>45 years) of *Pinus tabulaeformis* stands in southeast Loess Plateau, China, may be associated with decreasing plant species richness (Table 2), proportion of medium to large trees (Fig. 6), dominant tree biomass (Fig. 5) and soil nutrients (Fig. 7). In addition, proportion of seedlings was more than 45% when stands were protected for 60 and 75 years (Fig. 5). Forest tending operations are required to weaken competition among seedlings for sunlight, nutrients, moisture and space at those periods. We suggest that it is possible, based on the findings above, to couple forest management policy without exacerbating the poverty of local people, through the promotion of measured forest indices as evidence-based support for forest protection and use. For this region, we suggest the optimum forest protection age of 45 years would encourage maximum plant diversity and productivity, while supporting the socio-economic conditions of the local population for sustainable land use.

## CONCLUSIONS

The present study has reported differences of plant diversity, changes in forest age structure and soil nutrients of *Pinus tabulaeformis* stands restoration in chronosequence on the southeast Loess Plateau, China. The richness of plant species significantly differed with age of forest protection, attenuating towards more even distribution with increasing age of forest protection. Sustainable forest protection not only hindered increased biomass of dominant trees, organic carbon content, available phosphorus and potassium in top soil, but it also abated proportion medium and big trees after protection of 45 years. Meanwhile, proportion of seedlings and young trees increased to aggravate competition of environmental resources. Our findings have practical implications. By using measured forest indices as evidence-based support for balancing forest management policy, ecological restoration and local economy development including sustainable timber harvesting, we conclude that the preliminary optimal age for forest protection in this area should be not more than 45 years. Forest tending operations have to be implemented thereafter.

## ACKNOWLEDGEMENTS

We thank Mr. Ren-he Wang, Ms. Fen-ling Zhang and other staff in Huanglongshan Forest Bureau, Yanan, Shaanxi, China for their valuable assistance. We also thank Dr. Diane Allen for her comments.

### Funding

This study was funded by the National Forest Management Basic Project (No. 1692016-07) from the State Forestry Administration of the People's Republic of China. The funders had no role in study design, data collection and analysis, decision to publish, or preparation of the manuscript.

### Grant Disclosures

The following grant information was disclosed by the authors:
National Forest Management Basic Project: 1692016-07.

### Competing Interests

The authors declare there are no competing interests.

### Author Contributions

- Lin Hou conceived and designed the experiments, performed the experiments, analyzed the data, wrote the paper, reviewed drafts of the paper.
- Sijia Hou analyzed the data, contributed reagents/materials/analysis tools, prepared figures and/or tables.

### Data Availability

 The raw data has been supplied as Supplementary File.

### Supplemental Information

Supplemental information for this article can be found online at http://dx.doi.org/10.7717/peerj.3764#supplemental-information.

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
