# Peer review of "How long should the fully hillside-closed forest protection be implemented on the Loess Plateau, Shaanxi, China?"

_PeerJ, doi:10.7717/peerj.3764_

## Round 0.1 · original submission · Major Revisions

In addition to the thorough comments made by referee, you need to make the following changes:

- How were the plots chosen? Randomly (and then you need to explain how) or as a convenience sample? A map would probably help to assess how close are the plots to each other. Remember that all your statistical analyses depend on assuming that your plots represent independent replicates. If the plots of a given age come all from the same area, they may reflect more the characteristics of that area than age.
- Provide exact P-values and not just p<0.05. As P-values are measure of evidence against the null hypothesis, values matter.
- l. 161-162: which graphic checks did you perform on the residuals? Constant variance, influential values (eg Cook's distance), normality using qq plots? There are many diagnostics available, you need to be more specific.
- Figure 2: provide 95% CI for evenness (also in Figure 1 for richness and 3 for biomass). 95% CI cannot be compared directly but they provide a crude way to assess uncertainties. Given that evenness is constrained to be between 0 and 1, you can consider a transformation and then back-transform CI to the original scale. On Figure 3, do you show also SD (as in figure 1)? That would imply a huge variation among plots.

·

Basic reporting

This is an informative article about natural regeneration and the impact of such forest regeneration practice on sloping landscape. The article is generally well written although I edited it for mostly language, knowing that the authors are not native English speakers. The table and figures are clear and support the article. It is an important article for forest conservation both in China and elsewhere in the world.

I have attached the edited text for the authors' attention.

Experimental design

The experimental design is acceptable and so resulted in the data that supports the results. I suggest the authors replace Fig 2 with two separate figures (Fig 2a and Fig 2b) such that Fig 2a is the current Fig 2 while Fig 2b is the current Fig 2 with straight trend line for tree, shrub and herb. This will clearly show the trend of evenness index with age.

Validity of the findings

No comment

Additional comments

An interesting work that adds credence to China's investment in large scale forest restoration and land conservation programs. Specific results about optimum age of forest protection is important for each region of China and this paper did just that for sloping land protection on Loess Plateau, Shaanxi, China

Reviewer 2 ·

Basic reporting

Hou et al. propose a manuscript dedicated to the protection of fully hillside-closed forest on the Loess Plateau. The topic is of interest with respect to the Natural Forest Protection Program sustained by the Chinese government. The prime interest of paper is to decipher the optimal time span for Pinus tabulaeformis stands. However, the result of this experiment can not answer the scientific questions well which the authors presented. So this paper needs to be improved.

Experimental design

Experiment design and measurement in section Materials and Methods is not clear, suggest the authors to make it more clear by adding some more details.

Validity of the findings

The topic is of interest with respect to the Natural Forest Protection Program sustained by the Chinese government.The prime interest of paper is to decipher the optimal time span for Pinus tabulaeformis stands. However, the result of this experiment can not answer the scientific questions well which the authors presented.

Additional comments

1.Experiment design and measurement in section Materials and Methods is not clear. How to choose and arrange the plot and sub-plot?
2.L145-148 What is the basis of the classification method? Cited reference or others? Please make it clear.
3.In discuss section, authors discussed that the changes of richness is due to some pioneer tree species and drought resistant shrubs and herbs, moreover, soil moisture is the key limiting factor on the Loess Plateau, but why not provide the data of soil moisture of each vegetation in the manuscript?
4.L238 mean of this sentence is confused, please make it clear.
5.The section of “The optimal age for the fully hillside-closed forest protection” should be discussed and stated more specific. Because the diversity, biomass and soil nutrients showed different trend with the increase of the age. The optimal age should be determined by different purpose. e.g. for diversity is 30, for biomass is 45, for soil nutrient is 60.
6.L309-310 According to the results of the chart, the meaning of this sentence is inaccurate, please review.

Reviewer 3 ·

Basic reporting

Authors: Hou et al.,
How long should the fully hillside-closed forest protection be implemented on the Loess Plateau, Shaanxi, China?
Ms. ID: PeerJ-16347v1

This paper mainly discusses the trends of the plant species diversity, tree biomass, age distribution of trees and soil properties with the increase of hillside-closed forest protection time in the Loess Plateau of Shaanxi, China. The population of Pinus tabulaeformis was selected as a case study. The protection year (ranging from 10 to 75 years) was considered as the independent variable for the study. The Shannon-Wiener and evenness indices were used to analyze the community diversity. The results show that the richness of Pinus tabulaeformis reaches the maxima at the protecting age of 30 years, while the biomass of Pinus tabulaeformis and some important soil components reach the maxima at the protecting age of 45 years. Also, the forest age structure varies significantly with the increase of protection time. Based on these results, the authors concluded that the optimal age for forest protection of the Loess Plateau should not be more than 45 years. This study is quite interesting since previously it was well-believed that the hillside-closed forest protection time should be as long as possible. This study seems to give new insight for relevant forest protection strategy. The non-monotonic trends of the forest properties are quite surprising. However, before it is acceptable to PeerJ, the organization of some parts should be revised. More details about the results and discussions should be provided. The data sources and measurements should be cited and clarified.

Therefore, I recommend that this paper can be acceptable for PeerJ after the authors revising the paper according to the following comments and suggestions.

Comments:
1. The organization of some parts are not logical. Here are the suggestions for revision:
i) Lines 79-93 should be combined as one paragraph;
ii) Lines 96-109 should be combined as one paragraph;
iii) Lines 111-120 should be combined as one paragraph.
2. Line 126: More background and introduction about the Shannon-Wiener and evenness indices should be provided. Relevant references should be cited.
3. Lines 167-168, Lines 220-223, Tables 1 and 2: The data sources should be cited;
4. Lines 308-311: The authors concluded that the optimal forest protection age of 45 years is the best. Though it looks quite reasonable from the results of plant richness and biomass, some soil properties look more unpredictable since they contain some fluctuating trends. It would be better if the authors can provide more reasons and details for how they concluded this optimal protection time.
5. Figures 1, 3 and 4: Where do the error bars come from? Details should be mentioned in the captions.
6. Figure 4: The Roman numerals (I-V) should be directly explained in the caption, though they have been mentioned in the main text.
7. Figure 5: There are some typos in the titles of the x-axis. Please double check. More details about each subfigure should be provided in the caption.
8. Conclusion (Lines 312-322): The Conclusion part is not informative. A brief summary about the results is missing.

Experimental design

In my opinion, the experimental design and field measurement are suitable in this article.

Validity of the findings

There is a technical question about the validity of the findings:

Since the range of forest protection year mentioned in this article is very wide (from 10 to 75 years). How did the authors exactly know the year of the forest restoration? Please explain this in the manuscript.

---

## Round 0.2 · accepted · Accept

Thanks for carefully revising the manuscript and answering the comments of the reviewers.

·

Basic reporting

The authors have addressed all problems identified in the initial reviews with regard to language, structure of article, etc.

Experimental design

The experimental design is original, and all problems identified in the initial reviews have been corrected

Validity of the findings

The data, statistical analyses and inferences drawn from those analyses are sound.

Additional comments

I commend the authors for strictly adhering to the comments of the reviewers and revising the paper accordingly.

Reviewer 2 ·

Basic reporting

Clear and unambiguous, professional English used throughout.

Experimental design

Original primary research within Aims and Scope of the journal.

Validity of the findings

Impact and novelty not assessed. Negative/inconclusive results accepted. Meaningful replication encouraged where rationale & benefit to literature is clearly stated.

Additional comments

I have also recognized that the revision of the paper has considerably improved compared with the first version. All the reviewers questions have been answered, my view is that the paper can be published.

Reviewer 3 ·

Basic reporting

This revised manuscript has addressed all my previous concerns. It looks much better than the previous version. Therefore, I recommend that this paper in its current form is acceptable for PeerJ.

Experimental design

no comment

Validity of the findings

no comment